# Effect of Sugar Beet Pulp on the Composition and Predicted Function of Equine Fecal Microbiota

**DOI:** 10.3390/biology12091254

**Published:** 2023-09-19

**Authors:** Tamara Ford, Zachary L. McAdams, Kile S. Townsend, Lynn M. Martin, Philip J. Johnson, Aaron C. Ericsson

**Affiliations:** 1College of Veterinary Medicine (CVM), University of Missouri (MU), Columbia, MO 65211, USA; 2Molecular Pathogenesis and Therapeutics (MPT) Program, University of Missouri (MU), Columbia, MO 65201, USA; 3Department of Veterinary Medicine and Surgery, College of Veterinary Medicine (CVM), University of Missouri (MU), Columbia, MO 65211, USA; 4MU Metagenomics Center, Department of Veterinary Pathobiology, College of Veterinary Medicine (CVM), University of Missouri (MU), Columbia, MO 65201, USA

**Keywords:** horse, equine, microbiota, sugar beet pulp, prebiotic

## Abstract

**Simple Summary:**

As hindgut fermenters, horses are reliant on the complex microbial communities residing in the digestive tract. The composition of the fecal microbiome is shaped by dietary factors among others, and common but potentially fatal gastrointestinal conditions, such as colic and colitis, are associated with changes in the fecal microbiome. In relation to health, certain bacteria in an equine hindgut utilize dietary fiber to produce short-chain fatty acids, and sugar beet pulp is a high-fiber feed supplement believed to act as a prebiotic and promote good equine gut health. The information regarding the influence of sugar beet pulp and other prebiotics on the fecal microbiome of horses is limited. Evidence of the effect of sugar beet pulp on the fecal microbiome might support its use as a routine preventive measure against colic and colitis.

**Abstract:**

The purpose of this study is to determine the effect of the partial replacement of dietary hay with sugar beet pulp (SBP) on the composition and predicted function of the fecal microbiota of healthy adult horses. Fecal samples were collected daily for 12 days from six adult horses after removal from pasture, including a five-day acclimation period, and a seven-day period following the introduction of SBP into their diet, and compared to six untreated horses over a comparable period. Fecal DNA was subjected to 16S rRNA amplicon sequencing and a longitudinal analysis was performed comparing the composition and predicted function. While no significant treatment-associated changes in the richness, alpha diversity, or beta diversity were detected, random forest regression identified several high-importance taxonomic features associated with change over time in horses receiving SBP. A similar analysis of the predicted functional pathways identified several high-importance pathways, including those involved in the production of L-methionine and butyrate. These data suggest that feeding SBP to healthy adult horses acutely increases the relative abundance of several Gram-positive taxa, including *Cellulosilyticum* sp., *Moryella* sp., and *Weissella* sp., and mitigates the predicted functional changes associated with removal from pasture. Large-scale studies are needed to assess the protective effect of SBP on the incidence of the gastrointestinal conditions of horses.

## 1. Introduction

Colic is a condition in horses that involves abdominal pain with numerous possible etiologies, including dietary change, impaction, parasitic infestation, stress, dental issues, and colitis, among others [1]. Between 3.5 and 10.6 cases of colic per 100 horses occur each year [2,3]. Some have reported that colic arises in one of four horses annually [3]. Depending on the severity, colic may be managed medically or require surgical correction, which has varying levels of success. Following lameness, colic is the costliest condition to the equine industry, with losses to death alone totaling USD 115 million in 1998 [4]. In 2019, the most recent study estimating the cost of veterinary treatment for colic, the average expense of medical and surgical treatments was GBP 1501.08 (USD 1919) and GBP 6436.80 (USD 8221), respectively [5].

Colitis is one of the top ten causes of large intestinal colic in horses, and it can also be a sequela of other colic etiologies [6]. While similar to colic with respect to having an expansive range of etiologies, a minority of colitis cases achieve a definitive diagnosis. Considering the prevalence of colic and colitis and their threat to equine health and industry, it is exceedingly important to identify beneficial prophylactic treatments. Colic and colitis commonly involve hindgut dysbiosis [7,8,9], suggesting the equine gut microbiome might represent a target for preventive measures via the administration of probiotic bacteria or feed additives serving as nutrients for beneficial bacteria already present in the gastrointestinal tract (i.e., prebiotics).

Butyrate is a short-chain fatty acid that mediates anti-inflammatory and anti-proliferative effects through its role as a histone deacetylase (HDAC) inhibitor [10] and serves as an energy source for colonocytes [11], allowing tight junctions to form properly in the colon [12,13]. Butyrate is produced by certain Gram-positive anaerobes in the gut of horses and other species through the fermentation of dietary fiber, and production can be enhanced by feeding a higher percentage of fiber in the diet [14,15,16,17]. Notably, there is a shift during colitis favoring Gram-negative Bacteroidota over the Gram-positive Bacillota containing the majority of butyrogenic genera [7]. Unmolassed sugar beet pulp (SBP) is a high-fiber and low-sugar feed component that is the by-product of processing and extracting sugar from sugar beets [18]. Sugar beet pulp is a widely available feed that has been used for centuries and found to be a palatable supplement for older horses with difficulties in masticating and/or digesting hay as it is a rich source of hemicellulose and is highly digestible [19]. This feed has been suggested to be a promising equine prebiotic with the potential to reduce the incidence levels of colic and colitis [20]. Such a prebiotic can serve to stabilize bacterial communities in the gut. We hypothesize that the partial replacement of Timothy grass hay with SBP mitigates the changes in the composition of the fecal microbiome associated with removal from pasture, and that compositional changes in horses receiving SBP favor the expansion of known butyrogenic taxa. To test this hypothesis, we characterized the gut microbiome of healthy adult horses pre- and post-supplementation of their diet with SBP, alongside the historical data of similarly treated horses not receiving SBP. Both cohorts comprise horses moving from the same pasture to the same teaching hospital, and a robust sampling schedule is used. Traditional metrics of microbiome community structure are compared, and random forest regression analyses are used to identify differentially abundant taxa over time. Lastly, bioinformatic tools are used to infer functional changes in the fecal microbiome associated with SBP supplementation.

## 2. Materials and Methods

### 2.1. Horses and Diet

The six horses receiving SBP in this study were from the University of Missouri (MU) teaching herd located at MU Middlebush Farm. All procedures were approved by MU IACUC (Protocol #EX-10172). The horses were removed from the pasture and transported to the MU Veterinary Health Center (VHC) in early June 2021 and started on a diet of Timothy grass hay (10 pounds twice daily). The horses were acclimated to the new environment and diet for a period of five days prior the addition of SBP. The diet was then changed in the SBP group such that 2.5 pounds of soaked unmolassed SBP was fed twice daily for 7 days, replacing 2.5 pounds of Timothy grass hay for each feeding schedule. Fecal samples were collected daily during the five-day acclimation and seven-day SBP diet periods. In the VHC, the horses were housed in stalls with cedar shaving bedding with free access to fresh water. The horses were walked by hand in the VHC daily. Control horses were from the same herd and were removed from the same pasture and transported to the MU VHC during summer of the preceding year. Control horses were maintained on the Timothy grass hay diet throughout the study. The demographics of horses in each group are provided in Appendix A.

### 2.2. Fecal Sample Collection and Processing

The fecal samples were collected between 7:00 and 9:00 a.m. from freshly evacuated fecal piles. The interior portion of the fecal ball was placed in a plastic vial, and the sealed vials were placed in a freezer (−20 °C) and later transferred to the laboratory once the sample collection was completed. DNA was extracted from the samples using the QIAamp PowerFecal Pro Kit (Qiagen, Germantown, MD, USA) per the manufacturer’s instructions, and utilizing a Qiagen TissueLyser II for 10 minutes at 30 Hz for a mechanical lysis instead of a vortex and adapter. The DNA was quantified fluorometrically using a Quant-iT BR dsDNA reagent kit (Thermo Fisher, Waltham, MA, USA) and a Qubit 2.0 fluorometer (Thermo Fisher). The samples were standardized to the same concentration and volume for library preparation at the MU Genomics Technology Core facility. Bacterial 16S rRNA amplicons were generated via V4 variable region amplification using dual-indexed primers (U515F/806R) flanked by standard Illumina adapter sequences, and the following parameters: 98 °C (3 min) + [98 °C (15 s) + 50 °C (30 s) + 72 °C (30 s)] × 25 cycles + 72 °C (7 min). The amplicons were pooled, mixed, purified using Axygen Axyprep MagPCR clean-up beads, washed several times with 80% ethanol, and resuspended in EB buffer (Qiagen). The final amplicon pool was assessed with an Advanced Analytical Fragment Analyzer automated electrophoresis system, quantified using Quant-iT HS dsDNA reagent kits, diluted, and sequenced as 2 × 250 bp paired-end reads using the Illumina MiSeq platform and V2 chemistry based on the manufacturer’s instructions.

### 2.3. Informatics

The 16S rRNA sequence processing method was performed within the Quantitative Insights into Microbial Ecology 2 (QIIME2) [21] framework v2021.8. Briefly, paired-end reads were trimmed of Illumina adapter and primer sequences using cutadapt [22]. The trimmed sequences were then denoised using the Dada2 [23] method. Forward and reverse reads were truncated to 150 bp before denoising into amplicon sequence variants (ASVs). Chimeras were removed using the consensus method and ASVs were filtered to between 249 and 257 bp in length. ASV feature tables were rarefied to 24,190 features per sample. Taxonomy was then assigned to each ASV using a sklearn algorithm [24] with the SILVA 138 16S rRNA 515F-806R reference database [25]. The functional capacity (i.e., metagenome) at the MetaCyc [26] pathway level was predicted using the Phylogenetic Investigation of Communities by Reconstruction of Unobserved States 2 (PICRUSt2) [27] v2.5.1 with default settings.

Microbial community analysis and data visualization were performed within the open-source R statistical software [28] v4.2.2. Sample richness (i.e., Chao-1 Index) and Shannon diversity were determined using the *microbiome* [29] and *vegan* [30,31] libraries, respectively. Beta diversity was assessed using the *vegan* library. Briefly, a distance matrix using Bray–Curtis distances was created using a quarter root-transformed ASV feature table. Principal coordinate analysis (PCoA) with a Cailliez correction was performed on the distance matrix using the *ape* [32] library. Longitudinal microbiome and metagenomic pathway analyses were performed using the feature volatility tool within the QIIME2 q2-longitudinal [21] plugin. Important features were determined using a random forest regression using the sample date as the longitudinal state. All codes used for the microbial community analysis and data visualization are located at https://github.com/ericsson-lab/equine_sugar_beet (accessed on 14 July 2023).

### 2.4. Statistical Analysis

The sample coverage between groups was compared using a student’s *t*-test. Alpha diversity metrics and intra-subject beta diversity were compared using a two-way analysis of variance (ANOVA) with the treatment group and sample date as main effects. Differences in microbial community structure were compared using the *adonis* function within the *vegan* library [30,31]. A two-way permutational analysis of variance (PERMANOVA) was performed using the treatment and day as the main effects with 9999 permutations. Pairwise comparisons with Benjamini–Hochberg corrections [33] were performed using *EcolUtils*. A *p* < 0.05 was considered significant.

## 3. Results

### 3.1. Richness and Alpha-Diversity over Time

All samples amplified and sequenced well with mean (±SD) sequencing depths of 124,336 (±15,024) and 163,517 (±25,691) reads per sample in the control and SBP horses, respectively (Appendix A). There was a substantial amount of inter-subject variability in the richness (Figure 1A) and Shannon diversity (Figure 1B). Two-way ANOVA indicated a significant difference between the SBP and control horses (*p* = 0.0003, F = 13.7) in richness; however, the lack of a significant change over time (*p* = 0.71, F = 0.7) or interaction between treatment and time (*p* = 0.74, F = 0.7) suggested this to be a random difference. No significant treatment or time-associated differences or interactions were detected in the Shannon diversity.

### 3.2. Beta Diversity over Time

Similarly, the visualization of beta diversity among the SBP and control horses suggested a difference in composition between groups, captured primarily by the first two principal coordinates (Figure 2). PERMANOVA confirmed that this was a significant difference (*p* < 0.001, F = 16.3). As this analysis compared the two groups across all time-points, we then tested for differences within the SBP group, comparing the samples collected before (days 1 to 5) or after (days 6 to 12) the dietary switch. While samples were clustered primarily by horse and there was little evidence of treatment or time-associated changes in the beta diversity in PCoA, PERMANOVA detected a significant (albeit more subtle) difference (*p* = 0.0012, F = 2.8).

For a more granular comparison of the changes across time controlling for inter-individual variability, we used q2-longitudinal. The distance (i.e., dissimilarity) between the baseline and all subsequent timepoints, and between each successive timepoint, were calculated and compared using two-way ANOVA. As expected, the distance between the baseline and each subsequent timepoint increased significantly over time (*p* = 4 × 10^−13^, F = 11.4), most prominently during the first several days (Figure 3A). However, no significant group-associated difference (*p* = 0.25, F = 1.4), or interaction between group and time (*p* = 1.0, F = 0.2), was detected. Pairwise comparisons of the timepoints revealed significant differences within each treatment group between days 2 and 4, and between day 2 and all days beyond day 4. No other significant pairwise differences were detected between any other days. Similarly, distances between successive timepoints and the previous day did not differ between groups (Figure 3B), but did differ over time (*p* = 0.0003, F = 13.7). Pairwise comparisons detected an overall significant difference between days 3 and 11, and no other differences. Collectively, these data suggest that SBP has minimal to no effect on the global richness, alpha diversity, or beta diversity of the healthy equine fecal microbiota.

### 3.3. Random Forest Regression

To identify specific taxa that may be influenced by SBP, we first assessed the phylum-level relative abundance over the twelve-day sampling period. Of the dominant phyla (relative abundance > 1%) in the control group, *Bacillota*, *Bacteroidota*, and *Verrucomicrobiota* significantly differed over time (Appendix A). No dominant phyla differed over time in the group fed with SBP (Appendix A). Furthermore, the *Bacillota*-to-*Bacteroidota* ratio (formerly *Firmicutes*-to-*Bacteroidetes* ratio) did not differ over time in either control or SBP-fed groups (Appendix A). The *Bacillota*-to-*Bacteroidota* ratio was significantly higher in the control horses compared to the SBP-fed horses; however, given that the ratio did not change after introducing SBP, this difference in the relative abundance of *Bacillota* and *Bacteroidota* was likely due to natural variations in the equine gut microbiome over extended periods of time.

Considering SBP did not influence the relative abundance of dominant phyla, we applied a random forest (RF) regression approach to identify the genus-level changes induced by diet. A longitudinal feature volatility analysis identified 31 genera of importance for explaining the compositional changes over time, with *Sutterella*, *Cellulosilyticum*, and *Alloprevotella* spp. (Figure 4A) contributing roughly 25% of total importance. The mean net change in the relative abundance (RA) of the identified taxa in each group was modest, and the most substantial changes in the control group were abrogated in horses receiving SBP (Figure 4B). A visualization of the RA of the top three taxa across time in each group revealed gradual reductions in the RA of Gram-negative *Sutterella* and *Alloprevotella* and an increase in the RA of *Cellulosilyticum* in the SBP group, beginning after the diet switch (Figure 4C–E). Similarly, an examination of the subsequent three taxa in order of importance revealed a reduction in Gram-negative *Colidextribacter* (Appendix A) and expansions in the response to the SBP of Gram-positive *Moryella* sp. and *Weissella* sp. (Appendix A).

### 3.4. Predicted Functional Profiling

Changes or differences in global beta -diversity or the RA of specific taxa over time are not necessarily accompanied by functional changes. To assess the predicted function in SBP horses over time, in comparison to untreated horses, PICRUSt2 [27] was used, followed by the same RF regression and longitudinal feature volatility analysis employed with the taxonomic data. Of the 419 total pathways identified, 367 contributed some level of importance. The top 25 pathways, contributing a total of 42.2% of the overall predicted functional change, are shown in Figure 5A. A visualization of the mean net change in the RA of the top 25 pathways revealed an interesting pattern where predicted functional changes in the control group, whether positive or negative, were greater in the control group than in horses receiving SBP, with only two exceptions (PWY-7003, glycerol degradation to butanol; and HSERMETANA, L-methionine biosynthesis III) (Figure 5B). The four most important pathways included PWY-5971 (palmitate biosynthesis II; type-II fatty acid synthase), PWY-7003, PWY-7221 (guanosine ribonucleotides de novo biosynthesis), and HEXITOLDEGSUPER-PWY (super pathway of hexitol degradation) (Figure 5C–F). We noted that the net change in the predicted pathway PWY-7003 represented an artifact of the baseline RA in each group, as the changes in each group beyond day 6 favored horses receiving SBP, whereas the RA of this pathway in the control horses remained very low (Figure 5D). The fifth most important pathway (CENTFERM, pyruvate fermentation to butanoate) was also related to the production of the short-chain fatty acid (SCFA) butyrate; however, the positive change over time was greater in the control group than the SBP group. Collectively, these data suggest that the predicted functional changes occurring as a result of moving from pasture to barn are largely mitigated in the group receiving SBP. Moreover, certain taxa, including *Cellulosilyticum*, appear to proliferate in response to SBP, and the dominant predicted functional changes associated with SBP include increased L-methionine biosynthesis and production of butanol as a metabolic product of glycerol degradation, and the reduced fermentation of pyruvate to butanoate (butyric acid).

## 4. Discussion

While the benefits of prebiotics and probiotics on the health of humans [34,35] and livestock [36,37,38] are well-appreciated, their benefits to equine health are much less clear [39,40,41]. The data supporting the use of probiotics in horses are limited. Considering the relatively larger size of the equine gastrointestinal tract, it is difficult to imagine probiotic organisms surviving gastrointestinal transit and contributing significantly to the hindgut microbiome. Prebiotics, such as SBP, on the other hand represent a dietary fiber source allowing the continual inoculation of the existing microbiome. The anaerobes in the hindgut capable of degrading these carbohydrates express a diverse assortment of carbohydrate-active enzymes (CAZymes) [42], comprising glycoside hydrolases, polysaccharide lyases, and carbohydrate esterases (among other auxiliary and carbohydrate-binding modules), allowing for the selective hydrolysis or non-hydrolytic cleavage of specific glycoside bonds present in a given prebiotic [43,44]. Ultimately, the combination of oligosaccharides present in the diet (manipulated via prebiotics) and CAZymes present in the microbiome determine the production of acetate (C2), propionate (C3), and butyrate (C4) SCFAs.

This is a dynamic process with primary degraders of diverse taxonomies (e.g., *Akkermansia*, *Bacteroides*, *Bifidobacterium*, *Ruminococcus*) hydrolyzing oligosaccharides into monosaccharides, acetate, and propionate, and secondary degraders primarily within the *Bacillota* (*Faecalibacterium*, *Eubacterium*, *Roseburia*, and many others) converting monosaccharides and acetate to butyrate and other fatty acids [45,46,47,48,49]. Sugar beet pulp is a rich source of several oligo- and monosaccharides, including mannose, rhamnose, glucuronic and galacturonic acids; glucose; galactose; arabinose; and fucose [50]. Previous work on pigs demonstrated the minimal effects of SBP supplementation on the commensal gut microbiome [51], or the selective enrichment of *Eubacterium ruminantium* in feces following 12 weeks of supplementation [52]. The effect of SBP on the rumen microbiome of beef calves was negligible [53]; however, the fecal microbiome of dairy cows revealed the selective enrichment of *Fibrobacter* sp. [54] The data from the current study suggest that the partial replacement of Timothy grass hay with unmolassed SBP favors the expansion of select bacteria in the fecal microbiome of horses, including several Gram-positive members of the *Bacillota* known to possess at least one of the pathways for butyrate production, including *Cellulosilyticum* [55], *Moryella* [56], and *Weissella* [57].

As the name implies, the genus *Cellulosilyticum* (fam. *Lachnospiraceae*) is capable of degrading cellulose and other oligosaccharides [55,58]. Previous studies have demonstrated the positive effect of prebiotics on the RA of *Cellulosilyticum* within the gut. *Cellulosilyticum* was among the few taxa enriched in the feces of thoroughbred racehorses receiving antibiotics and supplemented with a commercially available synbiotic (combined pre- and probiotic), relative to horses receiving antibiotics alone [59]. Similarly, in weaning-age pigs maintained in poor sanitary conditions, *Cellulosilyticum* was enriched in pigs receiving low-dose fermentable oligosaccharides intended to enhance the fermentability of other fibers present in the diet [60]. *Moryella* (fam. *Lachnospiraceae*) is capable of producing indole, lactic acid, acetate, and butyrate [56]; however, its response to dietary manipulation and prebiotics is not well-characterized. *Weissella* is closely related to *Lactobacillus* and produces high levels of lactic and acetic acids [61]. The purported health benefits of *Weissella* [62] are largely attributed to its ability to degrade several prebiotic oligosaccharides [63,64], and produce antimicrobial exopolysaccharides [65,66,67] favoring the growth of other probiotic species [68,69].

The horses used in the current study were all healthy adults. The analyses of colitic or diarrheic horses revealed similar changes in the fecal microbiome of affected horses relative to the healthy controls. Seminal work by Costa et al. demonstrated a reduced abundance of *Bacillota* with replacement by *Bacteroidota* in horses with clinical colitis [7]. More recent work based on a population of 55 colitic horses and 36 healthy controls observed similar reductions in the dominant *Bacillota* families *Ruminococcaceae* and *Lachnospiraceae* (including *Faecalibacterium*) in horses with colitis [70]. Horses with diarrhea showed similar changes at the phylum level and significantly reduced RA values of several SCFA-producing genera, including *Weissella* [71], suggesting that these dysbiotic changes can occur independent of inflammation. SCFAs are essential carbon sources for colonocytes [11], and butyrate in particular regulates cellular proliferation and immune responses via histone deacetylase (HDAC) inhibition [10] and tight junction permeability [13]. It is thus likely that the changes in the microbiome associated with diarrhea or colitis serve to exacerbate inflammation, and we speculate that supplementation with SBP or other prebiotic sources might prevent the occurrence of adverse gastrointestinal events in horses.

The predicted functional analysis provided unexpected but intriguing results. The differences between SBP and control horses in KEGG pathways were minimal, ostensibly due to the limited number of taxonomic differences. However, a comparison of the changes occurring in the control horses over time following their removal from pasture to the changes occurring in horses supplemented with SBP demonstrated a clear pattern suggesting the protective effect of SBP on changes in community function due to an abrupt diet change. Similar to the recent work showing the protective effects of synbiotics against concurrent antibiotic treatment [51], perhaps supplementations with SBP prior to and during diet changes or other stressors (e.g., travel) might provide a useful preventive measure.

The limitations to the current study included the relatively small sample size and short period of dietary challenge. Concern regarding these limitations was mitigated by the fact that significant differences were detected, nonetheless. As has been highlighted by others [72], the gut microbiome of horses can also be affected by other factors, including age, breed, geographic location, time of year, and technical details, all of which can create difficulties when comparing the results of similar studies or comparing the data between the control and experimental horses, if the data are not collected within the same period of time. For this reason, each horse served as its own control in our analysis, and a robust sampling schedule was employed.

## 5. Conclusions

The introduction of SBP into the diet of healthy horses results in an increased relative abundance of a select group of Bacillota, including the genera *Cellulosilyticum*, *Moryella*, and *Weissella*, with no observable effects on the overall alpha- or beta-diversity values. No specific microbial functions were predicted to be altered due to the supplementation with SBP; however, the functional changes in the microbiome associated with removal from pasture were collectively mitigated. Ultimately, the consistent increases in butyrate producers in the fecal samples of horses receiving SBP suggest that the partial replacement of grass hay with SBP increases the SCFA production capacity, in theory promoting overall gut wellness. The outcomes of this study can be utilized to improve management practices, reduce the incidence of large intestinal dysbiosis, and decrease emergency health issues in horses.

## Figures and Tables

**Figure 1 biology-12-01254-f001:**
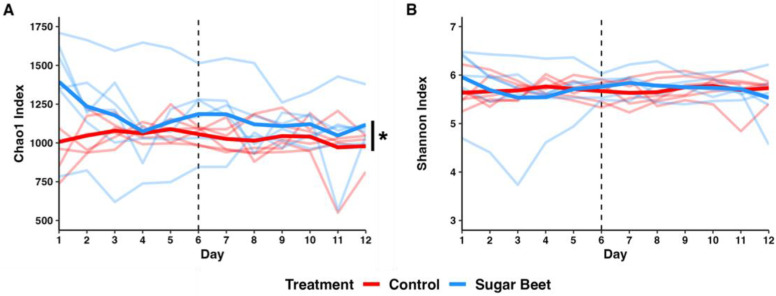
Spaghetti plots depicting (**A**) Chao1 richness and (**B**) Shannon diversity across time. Light lines depict individual horses; dark lines depict mean value across sampled timepoints. Dotted vertical line indicates diet change for SBP group on day 6 (D6). * *p* < 0.001, two-way ANOVA.

**Figure 2 biology-12-01254-f002:**
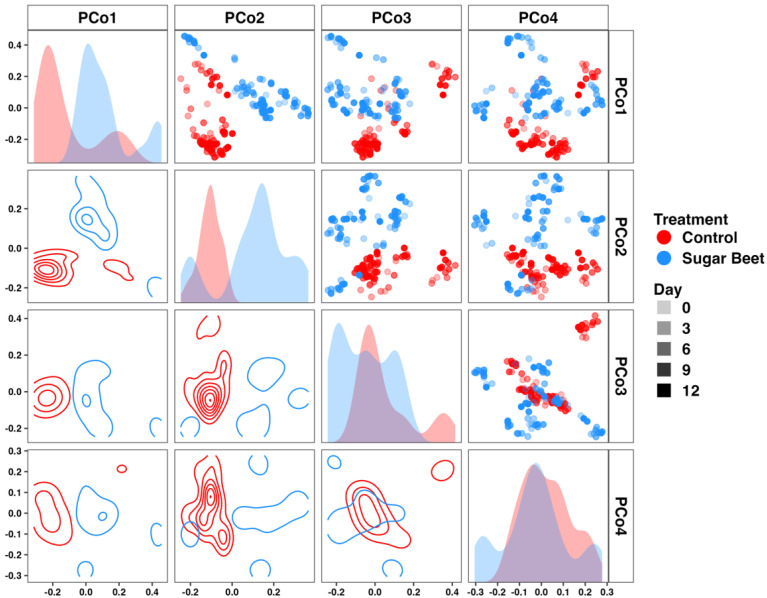
Matrix depicting beta diversity along the first four principal coordinates using Bray–Curtis distances. Dot plots depict individual samples along two axes. Opacity indicates day of study. Density plots depict sample distributions along two axes. Histograms depict sample distributions along a single axis.

**Figure 3 biology-12-01254-f003:**
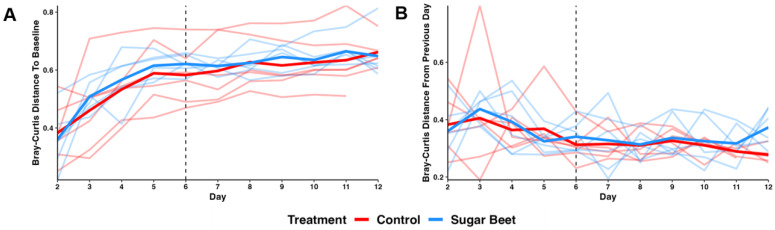
Spaghetti plots depicting (**A**) Bray–Curtis distance (i.e., dissimilarity) from each successive day to baseline (day 1), and (**B**) distance from each successive day to the previous day. Light lines depict individual horses; dark lines depict average distance of groups. Dotted vertical line indicates diet change for SBP group on day 6 (D6).

**Figure 4 biology-12-01254-f004:**
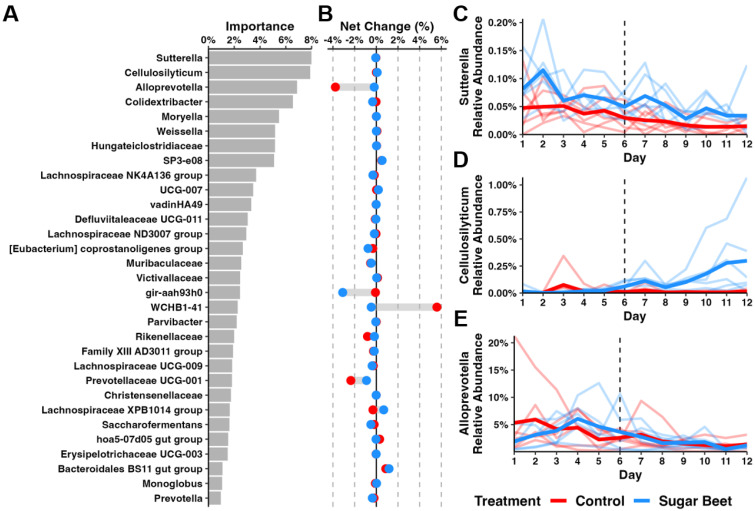
Longitudinal feature volatility analysis of genera detected across time. Bar plot showing (**A**) 31 important genera identified using a random forest approach. Dumbbell plots depicting (**B**) average net change in genera relative abundance. Spaghetti plots depicting relative abundances of (**C**) *Sutterella*, (**D**) *Cellulosilyticum*, and (**E**) *Alloprevotella* across time. Light lines depict individual horses; dark lines depict average relative abundance of groups. Dotted vertical line indicates diet change for SBP group on day 6 (D6).

**Figure 5 biology-12-01254-f005:**
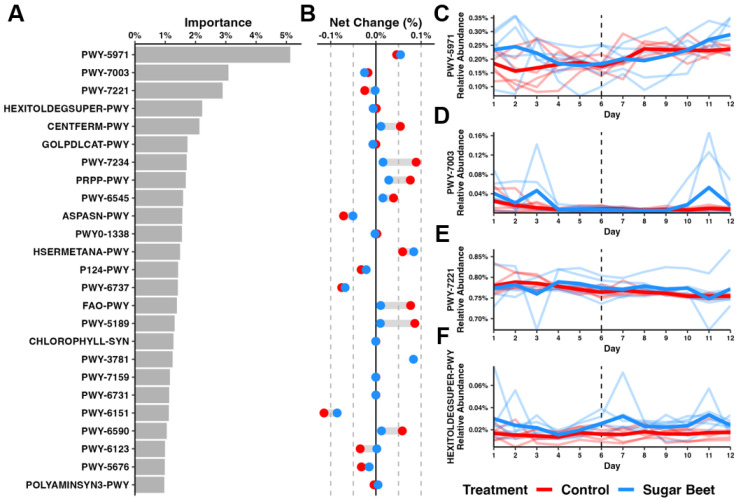
Longitudinal feature volatility analysis of PICRUSt2-predicted functional capacity across time. Bar plot showing (**A**) the top 25 most important pathways identified using a random forest approach. (**B**) Dumbbell plots depicting average net change in pathway relative abundance. Spaghetti plots depicting (**C**) PWY-5971, (**D**) PWY-7003, (**E**) PWY-7221, and (**F**) HEXITOLDEGSUPER-PWY pathway relative abundances across time. Light lines depict individual horses. Dark lines depict average relative abundance of groups. Dotted line indicates diet change for SBP group on day 6 (D6).

## Data Availability

All sequencing data described herein are deposited in the National Center for Biotechnology Informatics (NCBI) Sequence Read Archive (SRA) under BioProject ID PRJNA679208 (control horses) and PRJNA922475 (SBP group).

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
