# Peer review of "Effect of Sugar Beet Pulp on the Composition and Predicted Function of Equine Fecal Microbiota"

_biology, 2023, doi:10.3390/biology12091254_

Round 1

Reviewer 1 Report

General comments:

An interesting paper on the effect of beet pulp on the composition of the intestinal microbiota of horses. The authors themselves critically stated that the biggest limitation of the investigation was the small group of animals and the short time of the study. I agree with them, the investigation should be expanded to include long-term effects. In the context of SCFAs production, in addition to advanced biostatistical methods, one can benefit from additional simple analysis of SCFAs in fecal samples using high-performance liquid chromatography (15. Torii T., Kanemitsu K., Wada T., Itoh S., Kinugawa K., Hagiwara A. Measurement of short-chain fatty acids in human faeces using high-performance liquid chromatography: Specimen stability. Ann. Clin. Biochem. 2010;47:447-452. doi: 10.1258/acb.2010.010047). is a note of caution in the context of conducting further investigations in this area.

The authors focused on a highly developed biostatistical analysis which seems interesting, especially considering the Phylogenetic Investigation of Communities by Reconstruction of Unobserved States, however, a simple analysis of the fecal phyla present in particular phyla would be worth considering - this will allow additional reference of the results to literature data on the equine microbiome.

Material and method:

91: add information about horses (age, breed, gender) for both groups

96-97: It is described that in the investigation group, samples were taken for 5 days of the post-acclimation period. What about the samples after the start of beet pulp administration? And in what pattern were samples taken in control horses? please clarify.

Were the samples from the control group investigated after collection, or stored in the freezer until the collective investigation of the samples?

101-104: The way the methodology was established is a certain limitation of the investigation - the control group was studied a year earlier. Despite the use of the same type of hay, its composition in particular years may vary, causing an impact on changes in the fecal microbiome of horses.

150: Please justify the use of random forest approach in the analysis of relative frequency of genera

Results:

206: And what about the relative frequency at the phylum level (most often discussed in the context of horses) - has the relative frequency of any changed significantly? has the Firmicutes: Bacteroides ratio changed? In my opinion, as the first step in analyzing the results (after the alpha and beta diversity description) phylum level is very important to discuss.

Author Response

  1. An interesting paper on the effect of beet pulp on the composition of the intestinal microbiota of horses. The authors themselves critically stated that the biggest limitation of the investigation was the small group of animals and the short time of the study. I agree with them, the investigation should be expanded to include long-term effects. In the context of SCFAs production, in addition to advanced biostatistical methods, one can benefit from additional simple analysis of SCFAs in fecal samples using high-performance liquid chromatography (15. Torii T., Kanemitsu K., Wada T., Itoh S., Kinugawa K., Hagiwara A. Measurement of short-chain fatty acids in human faeces using high-performance liquid chromatography: Specimen stability. Ann. Clin. Biochem. 2010;47:447-452. doi: 10.1258/acb.2010.010047). is a note of caution in the context of conducting further investigations in this area. 

Response: We appreciate the Reviewer’s suggestion and agree that future investigation of a sugar beet pulp diet would benefit from SCFA analysis. While that type of analysis was outside the scope of the current project, we continue to collect samples and seek funding to pursue those types of analyses.

  1. The authors focused on a highly developed biostatistical analysis which seems interesting, especially considering the Phylogenetic Investigation of Communities by Reconstruction of Unobserved States, however, a simple analysis of the fecal phyla present in particular phyla would be worth considering - this will allow additional reference of the results to literature data on the equine microbiome.

Response: We thank the Reviewer for acknowledging our innovative bioinformatic approach. PICRUSt2, in particular, is a valuable tool to predict microbial metabolic pathways influenced by SBP that may be investigated in future work. We also agree with the Reviewer’s suggestion to include phylum-level relative abundance, and have added a supplementary figure illustrating the major (relative abundance > 1%) phyla detected in horses receiving control and sugar beet pulp diet (Figure S2A-B). These data indicate no change in dominant phyla relative abundance following introduction of a SBP diet.

  1. Material and method:

91: add information about horses (age, breed, gender) for both groups

Response: We appreciate the Reviewer’s suggestion and have added Table S1, containing the requested data from horses in both groups.  This Table is referenced in line 104 of the revised manuscript.

  1. 96-97: It is described that in the investigation group, samples were taken for 5 days of the post-acclimation period. What about the samples after the start of beet pulp administration? And in what pattern were samples taken in control horses? please clarify.

Response: We appreciate the reviewer’s keen review of our methods and acknowledge these details were unclear. To that end, we have revised the text (Lines 96-101) to clearly describe the sampling protocol which included daily collections five days prior to and seven days following the addition of SBP to the diet (12 total samples per subject).

  1. Were the samples from the control group investigated after collection, or stored in the freezer until the collective investigation of the samples?

Response: The control samples were collected and processed one year prior to those collected from the sugar beet pulp group, however, samples from both groups were stored in a freezer and processed within a comparable time frame using the same fecal DNA extraction kits.

  1. 101-104: The way the methodology was established is a certain limitation of the investigation - the control group was studied a year earlier. Despite the use of the same type of hay, its composition in particular years may vary, causing an impact on changes in the fecal microbiome of horses.

Response: The reviewer has highlighted an important limitation of the current study. We acknowledge that the nutritional composition of the provided hay may have differed between sampling periods, and the observed differences in alpha and beta diversity and phylum-level relative abundance are likely explained by long-term temporal variation in the equine gut microbiome based on differences in forage quality and other environmental factors from year to year. Recognizing the high degree of intra-subject variability in horses in general, our goal was to identify changes in diversity, composition, and predictive function within horses following introduction of SBP. The control horses provide a measure of the degree of change expected in untreated horses, obviating concerns regarding the baseline differences between groups.

  1. 150: Please justify the use of random forest approach in the analysis of relative frequency of genera

Response: We elected to use a random forest approach to identify relevant genera that were predictive of shifts in taxonomic composition over time for multiple reasons. First, our study evaluated two groups of horses with large inherit taxonomic diversity. Secondly, our sample size was relatively small compared to the number of timepoints in the longitudinal dataset. Finally, the group receiving SBP supplementation did not consume SBP until six days into sampling. These reasons collectively deterred us from pursuing a traditional differential abundance testing approach to identify relevant genera. The random forest approach allowed us to identify important genera that change in response to the SBP diet that may be leveraged in future studies.

  1. 206: And what about the relative frequency at the phylum level (most often discussed in the context of horses) - has the relative frequency of any changed significantly? has the Firmicutes: Bacteroides ratio changed? In my opinion, as the first step in analyzing the results (after the alpha and beta diversity description) phylum level is very important to discuss.

Response: As describe above, we have incorporated an additional analysis of the dominant phyla within control and SBP groups (Figure S2A-B). We have also determined the Bacillota-to-Bacteroidota (formerly Firmicutes-to-Bacteroides) ratio (Figure S2C). While a significant difference in the Bacillota-to-Bacteroidota ratio between Control and SBP groups was observed, this ratio did not differ across time. Given that the introduction of SBP did not influence this ratio, we elected to perform the random forest regression analysis at a more granular taxonomic level.

We would like to thank the Reviewer again for their careful consideration of this work. 

Reviewer 2 Report

Thank you for submitting the manuscript entitled " Effect of sugar beet pulp on the composition and predicted function of the equine fecal microbiota" . The manuscript is well written and an important topic in feeding horses.

Comments: do you have the nutritional analysis of the hay and beet pulp? Please include.

Regards 

Author Response

Thank you for submitting the manuscript entitled "Effect of sugar beet pulp on the composition and predicted function of the equine fecal microbiota" . The manuscript is well written and an important topic in feeding horses.

Comments: do you have the nutritional analysis of the hay and beet pulp? Please include.

Response: We appreciate the Reviewer’s suggestion, but unfortunately do not have a detailed nutritional analysis of the feed or sugar beet pulp (SBP) provided to the horses.  We have done our best to denote the type of hay provided (Timothy grass) and amount of SBP, with the latter being purchased from a local feed provider. We recognize this as a limitation of the study and plan to incorporate such analyses in future work.

Reviewer 3 Report

Dear Editor,

Many thanks for inviting me to review this paper. 

I believe this study aligns with the scope of the journal.

 Large-scale studies are needed to assess the protective effect of SBP on incidence of gastrointestinal conditions of horses.

Author Response

I believe this study aligns with the scope of the journal. Large-scale studies are needed to assess the protective effect of SBP on incidence of gastrointestinal conditions of horses.

Response: We appreciate the favorable review and agree that large-scale long-term studies incorporating colic incidence and other clinical parameters are needed.

Reviewer 4 Report

It was a very clear, nice, and interesting article to read; please consider the following edits. Thank you

Please re-check and cite all the statements from the previously published literature throughout the manuscript.

Line 63-65: Please cite the statement.

Line 70-88: Please cite the statements from the previously published literature.

Same for lines 330-335. Please cite the statement.

The Authors adequately point out limitations. Nice!

Author Response

It was a very clear, nice, and interesting article to read; please consider the following edits. Thank you

Please re-check and cite all the statements from the previously published literature throughout the manuscript.

Line 63-65: Please cite the statement.

Response: We appreciate eh Reviewer’s comments and suggestions.  We have added references regarding the effects of butyrate on histone deacetylase (HDAC) function and colonocyte health.

Line 70-88: Please cite the statements from the previously published literature.

Response: Again, we have added references to the relevant section.

Same for lines 330-335. Please cite the statement.

Response:  We have added salient references here as well.

The Authors adequately point out limitations. Nice!

Reviewer 5 Report

Congratulations to the authors for their work in this area. I have some concerns about this paper I would like to see addressed

1. please explain further the methodology of the control horses. Line 103 says that the control horses were examined in the preceeding year. Please expand and make this clearer in the methodology. Please then justify this decision in the discussion as to how this is an adequate control and how this allows for differences in hay production, climate, temperature and condition of the horses.  There is no information in the results that discuss this and to ensure if horses were examined a year apart that they and the feed was consistent for comparison. 

2. Please justify the decision in the methods to take the manure over a 2 hour period and if this was a limitation of the study. Rectal sampling will ensure that there is no potential for manure to be at a different temperature than the horse. This would allow more consistency and  reduce the risk of bacterial  population  change once the manure is passed from the horse to the ground.

3. I found the legends to the figures to be confusing and suggest clarification.  I would like clarification in the method or alternate wording of the legend.  Please explain further what is treatment and what is diet. this would then be easier evaluate on the graphs where there is treatment red control blue sugar beet. It is confusing as the method reads as if there is control vs sugar beet and so what is then treatment and what is diet. please make this clearer

Author Response

Congratulations to the authors for their work in this area. I have some concerns about this paper I would like to see addressed

  1. please explain further the methodology of the control horses. Line 103 says that the control horses were examined in the preceding year. Please expand and make this clearer in the methodology. Please then justify this decision in the discussion as to how this is an adequate control and how this allows for differences in hay production, climate, temperature and condition of the horses.  There is no information in the results that discuss this and to ensure if horses were examined a year apart that they and the feed was consistent for comparison.

Response:  The control group of horses was selected primarily based on the feasibility of sampling horses from the same herd at the University of Missouri Veterinary Hospital. Both groups were healthy, adult horses from the same herd allowing for us to control for most husbandry factors prior to transport to the Veterinary Hospital. We acknowledge that the nutritional content of the hay may vary between seasons, however, the primary focus of our study was to identify changes in diversity and taxonomic composition over time within a group rather than between groups.

  1. Please justify the decision in the methods to take the manure over a 2 hour period and if this was a limitation of the study. Rectal sampling will ensure that there is no potential for manure to be at a different temperature than the horse. This would allow more consistency and reduce the risk of bacterial population  change once the manure is passed from the horse to the ground.

Response: We believe that this is not a limitation of our study. The two-hour period was the approximate time frame in which samples were collected each day over the course of the twelve-day study. This window coincided with daily stall checks which will slightly vary between days. If a fecal sample were to be collected after two hours at room temperature, however, we do not expect major changes in microbial richness, diversity, or composition to have occurred. Multiple groups have described that major shifts in diversity and composition in equine fecal samples at ambient temperature do not occur for at least six hours following defecation1,2. Our sampling protocol was well within this six-hour window.

  1. I found the legends to the figures to be confusing and suggest clarification.  I would like clarification in the method or alternate wording of the legend.  Please explain further what is treatment and what is diet. this would then be easier evaluate on the graphs where there is treatment red control blue sugar beet. It is confusing as the method reads as if there is control vs sugar beet and so what is then treatment and what is diet. please make this clearer

Response: We appreciate your careful review of the figure legends. We have updated the figure to which we believe the reviewer is referring (Figure 2). Shape had been included to denote when the SBP group started receiving SBP, however, we have clarified this in the figure.

References

  1. Beckers, K. F., Schulz, C. J. & Childers, G. W. Rapid regrowth and detection of microbial contaminants in equine fecal microbiome samples. PLoS ONE 12, e0187044 (2017).
  2. Bustamante, M. M. de, Plummer, C., MacNicol, J. & Gomez, D. Impact of Ambient Temperature Sample Storage on the Equine Fecal Microbiota. Animals 11, 819 (2021).